# IFIT3 and IFIT5 Play Potential Roles in Innate Immune Response of Porcine Pulmonary Microvascular Endothelial Cells to Highly Pathogenic Porcine Reproductive and Respiratory Syndrome Virus

**DOI:** 10.3390/v14091919

**Published:** 2022-08-30

**Authors:** Yanmei Wu, Xiaoxiao Song, Defeng Cui, Tao Zhang

**Affiliations:** Beijing Key Laboratory of Traditional Chinese Veterinary Medicine, Animal Science and Technology College, Beijing University of Agriculture, Beijing 102206, China

**Keywords:** HP-PRRSV, MVECs, interferon-inducible protein, astragalus polysaccharide, interferon, autophagosome

## Abstract

Our previous study has demonstrated that porcine pulmonary microvascular endothelial cells (MVECs) are susceptible to highly pathogenic porcine reproductive and respiratory syndrome virus (HP-PRRSV). The innate immune response of MVECs infected with HP-PRRSV would play important roles in controlling virus proliferation, resisting cellular injury, and preventing the virus from spreading to other tissues and organs. Type I interferon is one of the most effective antiviral cytokines in the innate immune response, and interferon-induced proteins with tetratricopeptide repeats (IFITs) are members of interferon-stimulated genes induced by viruses and other pathogens, which are crucial in inhibiting virus proliferation and regulating the innate immune response. However, their effects on HP-PRRSV-induced innate immunity in porcine pulmonary MVECs remain unclear. Here, the roles of IFITs in porcine pulmonary MVECs infected with the HP-PRRSV HN strain were investigated, and the effects of astragalus polysaccharides (APS), a widely used traditional Chinese herbal ingredient with the immunopotentiating effect, on them were studied. The results showed that more autophagosomes were observed in HP-PRRSV-infected MVECs, and the expression of *IFN-α*, *IFIT3*, and *IFIT5* decreased or increased at different time points after infection. When silencing the genes of *IFIT3* or *IFIT5*, the HP-PRRSV replication in MVECs was significantly increased. The expression of *IFIT3* and *IFIT5* could be upregulated by APS, whose inhibitory effects on the HP-PRRSV replication significantly declined when the genes of *IFIT3* or *IFIT5* were silenced. The results suggest that IFIT3 and IFIT5 play an important role in inhibiting the HP-PRRSV replication in porcine pulmonary MVECs, and APS suppress the multiplication of HP-PRRSV by upregulating their expression.

## 1. Introduction

Porcine reproductive and respiratory syndrome virus (PRRSV) is an important pathogen that seriously jeopardizes the pig industry, mainly causing severe respiratory symptoms in infected piglets. Particularly, the highly pathogenic PRRSV (HP-PRRSV), which emerged in China in 2006 and now mainly occurs in Southeast Asian countries, caused more severe clinical symptoms and higher mortality [1]. Massive multiplication of HP-PRRSV in the piglet lung would cause severe pathological damage, and then they would spread with the blood flow to result in systemic infection, in which pulmonary microvascular endothelial cells (MVECs) play a key role. Our previous study demonstrated that porcine pulmonary MVECs are susceptible to HP-PRRSV [2]. It has been documented that MVECs play an important role in both innate and acquired immunity [3], and especially their innate response is important to control viral multiplication and spreading. Therefore, the microvascular endothelial dysfunction will not only cause various pathological injuries to the lung tissue but also lead to failure of the important defensive barrier. In the innate immune response, type I interferons (IFN-α/β) are the most effective antiviral cytokines and resist the viral invasion mainly by triggering the expression of interferon-stimulated genes (ISGs), among which interferon-induced proteins with tetrapeptide repeats (IFITs) are an important family of antiviral proteins with high sensitivity to viral infection. IFITs can inhibit viral multiplication and resist virus infection, and their antiviral activities are getting more and more attention.

Type I interferons are induced in viruses-infected cells, then bind to their receptors to trigger a series of signaling pathways, and eventually produce a large number of interferon-stimulating factors, including IFITs [4]. On the other hand, mitochondria are the signal aggregation platform for inducing type I interferon production. The virus infection can give rise to mitochondrial autophagic degradation and then suppress the type I interferon response of host cells [5]. The IFITs family of most mammals includes four members: IFIT1/ISG56, IFIT2/ISG54, IFIT3/ISG60, and IFIT5/ISG58. It has been shown in the literature that the expression of IFIT1 and IFIT2 is significantly enhanced in the infection of DNA viruses, such as cytomegalovirus and adenovirus [6,7], and the altered expression of IFIT3 and IFIT5 occurred more frequently in the infection of RNA viruses, such as Hantavirus and influenza virus [8,9]. Zhang et al. demonstrated that overexpression of IFIT3 in Marc 145 cells reduced the replication of HP-PRRSV, while the gene silence of IFIT3 increased its infection [10]. Ma’s study on *IFIT5* using alveolar macrophages also had similar results, and he found that inhibiting or enhancing *IFIT5* expression would increase or reduce the HP-PRRSV replication [11]. Although the production of IFITs in PRRSV-infected cells has been reported previously, the roles of IFIT3 and IFIT5 in HP-PRRSV-infected porcine pulmonary MVECs remain unclear.

Astragalus polysaccharides (APS), the most widely used Chinese herbal ingredient, possesses the immunopotentiating effect in both research and clinical application. It is well known that APS can promote the production of type I interferon and also has a significant inhibitory effect on autophagy caused by oxidative damage [12,13]. In particular, it is increasingly paid attention to that immunoregulation effects of APS are studied from the perspective of MVECs [14,15]. In addition, the applications of APS to research in the prevention and treatment of porcine reproductive and respiratory syndrome and resistance to PRRSV have been reported frequently [16,17]. Therefore, we speculated that APS should be able to regulate the expression of IFITs in porcine pulmonary MVECs to resist HP-PRRSV.

In this study, to understand the roles of IFIT3 and IFIT5, porcine pulmonary MVECs in vitro were infected with the HP-PRRSV HN strain. Then, the ultrastructural changes of MVECs were analyzed, and the productions of type I interferon, IFIT3, and IFIT5 were detected. Furthermore, the effects of gene silencing of *IFIT3* and *IFIT5* on the HP-PRRSV multiplication in MVECs and on the anti-HP-PRRSV effect of APS were investigated. Our results provided experimental data to clarify the innate immune response of porcine pulmonary MVECs to HP-PRRSV and guided the application of APS in the prevention and treatment of PRRS.

## 2. Materials and Methods

### 2.1. Virus

The HP-PRRSV HN strain was kindly provided by Dr. Zhanzhong Zhao from the Chinese Academy of Agricultural Sciences and propagated and titrated on Marc-145 cells. After repeated freezing and thawing to lyse the cells and centrifuging to remove the cell debris, the virus culture medium was collected, whose viral titer was determined to be 10-6.3 TCID50/100 μL. The normal cultured Marc-145 cell lysate was collected in the same way as the control medium. Both viral medium and control medium were frozen at −80 °C for subsequent tests.

### 2.2. Isolation and Culture of Porcine Pulmonary MVECs

Porcine pulmonary MVECs were isolated from 10-day-old SPF *piglets* (purchased from Beijing SPF Pig Breeding Management Center), and all procedures were approved by the Animal Care and Protection Committee of Beijing University of Agriculture (No. BUA_ZT202001). According to previous methods [14,18], the primary cell culture was isolated and cultured from the lung marginal tissue by collagenase type II digestion and differential cell adhesion, and then MVECs were purified using immunomagnetic beads coated with anti-CD31 antibodies. After subculture for 24 h, cells were fixed with 4% paraformaldehyde to perform immunofluorescence staining of CD31.

All experiments were carried out on MVECs in passages 4–5. The cells were infected with HP-PRRSV by incubation in the virus medium for 1 h, or they were incubated in the control medium for 1 h as the control group. APS were used at the concentration of 100 μg/mL, which were added when the virus medium was replaced in the treatment group. Both groups were sampled at 12, 24, and 48 h post infection (hpi), respectively.

### 2.3. Immunofluorescence Staining

The indirect immunofluorescence staining was performed with reference to the previous literature [19]. In brief, the fixed cells were washed with PBS and blocked with 3% BSA at room temperature for 30 min. CD31 antibodies (Proteintech, 11265-1-AP) were added and incubated overnight at 4 °C. The negative control group replaced the antibody with PBS. After washing, FITC-labeled goat anti-rabbit IgG was added and incubated in the dark at room temperature for 2 h. Nuclei were counterstained with 2 μg/mL DAPI for 5 min. The cells were observed and photographed with a fluorescence microscope (IX71, Olympus, Tokyo, Japan).

### 2.4. Transmission Electron Microscopy (TEM)

MVECs were infected for 24 h, 48 h, and 72 h and fixed with 2.5% glutaraldehyde solution for transmission electron microscope analysis. The steps are as follows. The cells were post-fixed with 1% osmium tetroxide at 4 °C for 2 h and then dehydrated with 30%, 50%, 70%, 80%, 90%, 100%, 100% ethanol gradient and replaced with acetone twice for 10 min each time. They were immersed in SPI-Pon^TM^ 812 epoxy resin containing different amounts of acetone (75%, 50%, 25%, 0%). Finally, the cells were embedded in pure resin containing 1.5% BDMA and polymerized at 45 °C for 12 h and 60 °C for 48h. Ultrathin sections of 70 nm were prepared with Leica microtome (Leica EM UC6), stained with uranyl acetate and lead citrate, which were observed and photographed with the transmission electron microscope (Fei TECNAI spirit 120kv).

### 2.5. Quantitative Real-Time PCR (RT-PCR)

The total RNA was extracted from porcine pulmonary MVECs infected with HP-PRRSV or treated with APS by the Trizol method and reverse transcribed into cDNA using the cDNA synthesis Kit (Tsingke, tsk301). The target gene was amplified in a quantitative RT-PCR instrument (LightCycler^®^ 96 SW, LightCycler^®^, USA). Total reaction volume is 20 μL: 2 μL cDNA template, 2 μL primer, 10 μ L PowerUp ™ SYBR ™ Green Master Mix (Applied Biosystems ™, 00791640), 7 μL ddH_2_O. Reaction conditions: pre-denaturation at 94 °C for 5 min, denaturation at 94 °C for 15 s, annealing at 58 °C for 15 s, and extension at 72 °C for 45 s, 40 cycles. The relative expression of the target gene was calculated by the 2^−ΔΔCT^ method. Primers (Appendix A) were designed by primer premier 5.0 software (American Premier Biosoft) and synthesized by Shanghai Sangong Bioengineering Co., Ltd. All quantitative RT-PCR experiments were performed in triplicate.

### 2.6. Western Blotting

The total cellular protein was extracted with RIPA strong lysate. The protein samples were electrophoresed on 10% or 15% polyacrylamide gel and transferred to PVDF membranes, which were blocked with a 5% solution of non-fat milk at room temperature for 1.5 h. GAPDH antibody (Proteintech, 66009-1-Ig, 1:10,000), IFIT3 antibody (Santa Cruz, sc-393512, 1:500), IFIT5 antibody (Proteintech, 13378-1-ap, 1:1000), and PRRSV N antibody (BIOSs, bs-23941r, 1:1000) were added, respectively, and incubated overnight at 4 °C. After washing the PVDF membrane with TBST 4 times, HRP-labeled Goat anti-mouse IgG (Beyotime, a0216) or HRP-labeled Goat anti-rabbit IgG (Beyotime, a0208) solutions were added and incubated at room temperature for 1.5 h. According to the manufacturer’s instructions, the protein bands were visualized on the chemiluminescence imager (Shanghai Tianneng Technology Co., Ltd., Tanon-5200) using an ECL kit (Vazyme, E412), and the gray value of the protein bands was analyzed with ImageJ software.

### 2.7. Cell Transfection

MVECs were seeded in 6-well culture plates and transfected with Entranstertm-R4000 (Engreen, 4000-3) when growing to about 50% confluence. According to the manufacturer’s instructions, siRNA and entranstertm-r4000 were diluted with Opti-MEM medium (GIBCO, 31985070), and their diluents were mixed into the transfection complex. After standing at room temperature for 15 min, it was added to the cells. At the same time, 1.9 mL of DMEM containing 10% serum was added to make the final siRNA concentration of 140 pmol/mL. Nonspecific siRNA was used as control. After incubation for 12 h, the complete medium was replaced. The gene silencing efficiencies of *IFIT3* and *IFIT5* were detected after 36 h. The specific siRNAs of *IFIT3* and *IFIT5* were designed and synthesized by gene Pharma (China, Appendix A). The transfected cells were infected with HP-PRRSV or/and treated with APS for 24 h, and the expression of *IFIT3*, *IFIT5*, and PRRSV N was detected.

### 2.8. Statistical Analysis

All experiments were repeated three times, and data were presented as “mean ± standard deviation”. One-way ANOVA or Student’s *t*-test was performed using GraphPad Prism 8.0, and *p* < 0.05 and *p* < 0.01 were considered statistically significant or extremely significant.

## 3. Results

### 3.1. Characteristics of Porcine Pulmonary MVECs

The primary cell culture of porcine pulmonary MVECs was obtained by digestion with type II collagenase and differential adhesion, and high-purity porcine pulmonary MVECs were separated and purified by CD31-coated immunomagnetic beads. As shown in Figure 1A, many CD31 immunomagnetic beads were bound to each porcine pulmonary MVECs after isolation from the primary cell culture. The purified MVECs were polygons with one or more long protrusions (Figure 1B). After about 5 days of passage, they grew into a confluent monolayer and showed growth inhibition. They maintained good proliferation performance and subculturing state until passage 7 and then aged rapidly after passage 8. Immunofluorescence analysis showed that they were positive for CD31, and the positive cell rate was about 97% (Figure 1C).

### 3.2. Confirmation of HP-PRRSV Infection in Porcine Pulmonary MVECs

To understand the multiplication characteristic of HP-PRRSV in porcine pulmonary MVECs and the effect of its infection on MVECs, the gene expression of PRRSV N and ultrastructural changes of cells were analyzed at different time points after infection. The results showed that from 3 to 96 hpi, the PRRSV N gene could be detected in MVECs cells. Interestingly, its expression firstly decreased from 3 to 48 hpi and then increased from 72 hpi (Figure 2A). At the protein level, positive immunofluorescence staining for PRRSV N protein was observed in the HP-PRRSV-infected MVECs (Figure 2B). TEM analysis showed that the virus particles were seen in HP-PRRSV-infected MVECs (Figure 3B), and more autophagosomes were seen in the cell cytoplasm at 72 hpi (Figure 3C,D), which did not appear in normal control MVECs. The above results confirmed infection of the HP-PRRSV HN strain in porcine pulmonary MVECs, and the presence of many autophagosomes suggested that the inflammatory and immune responses were induced by the HP-PRRSV infection.

### 3.3. IFN-α Expression in HP-PRRSV-Infected Porcine Pulmonary MVECs

To know the effect of HP-PRRSV infection on the production of Type I interferon in porcine pulmonary MVECs, the gene expression of *IFN-α* at different time points was detected. The data indicated that the *IFN-α* expression in HP-PRRSV-infected MVECs firstly increased compared to that in normal control MVECs at 3 hpi and then underwent a period of decline from 6 to 24 hpi (Figure 4). However, at 48 and 72 hpi, its expression significantly increased again. The result showed that effects of HP-PRRSV infection on the *IFN-α* gene expression in porcine pulmonary MVECs were complicated.

### 3.4. Expression of IFIT3 and IFIT5 in HP-PRRSV-Infected Porcine Pulmonary MVECs

Considering the antiviral effects of interferons mainly depend on their induced downstream interferon-stimulated genes, the expression of *IFIT3* and *IFIT5* in HP-PRRSV-infected porcine pulmonary MVECs was detected. The results showed that the expression of *IFIT3* and *IFIT5* was changed in a similar manner at the mRNA and protein levels (Figure 5). At 3 hpi, their expression was temporarily increased in HP-PRRSV groups, and then at 6, 12, and 24 hpi, it was significantly lower than that in control groups. However, at 48 and 72 hpi, their expression was induced by the HP-PRRSV infection to increase again. Notably, when porcine pulmonary MVECs were infected with the HP-PRRSV HN strain, the change in expression of *IFIT3* and *IFIT5* was consistent with that of IFN-α, and they displayed the same kinetics pattern.

### 3.5. Effects of Silencing IFIT3 and IFIT5 Expression on the HP-PRRSV Multiplication

To reveal the roles of IFIT3 and IFIT5 in HP-PRRSV multiplication in porcine pulmonary MVECs, their genes were silenced. After transfection of the specific siRNA, the expression of *IFIT3* and *IFIT5* in porcine pulmonary MVECs was significantly decreased at both gene and protein levels (Figure 6). Furthermore, when one gene of *IFIT3* and *IFIT5* was silenced, the expression of the other gene was also significantly down-regulated, while its protein expression had no significant change.

IFITs are considered effective antiviral proteins, which have been verified in a variety of viruses and cells. In this study, after silencing the expression of *IFIT3* and *IFIT5*, the multiplication of HP-PRRSV in porcine pulmonary MVECs was measured. We found that PRRSV N expression at both gene and protein levels was significantly increased (Figure 7). Interestingly, the expression of PRRSV N increasing was more significant when silencing the *IFIT5* than when silencing the *IFIT3*. The data demonstrated that IFIT3 and IFIT5 played an important role in controlling the multiplication of HP-PRRSV in porcine pulmonary MVECs.

### 3.6. Effects of APS on the Expression of IFIT3 and IFIT5

Regulating the expression of immune cytokines is important for APS to improve the immune response. In order to reveal the role of IFITs in the regulation of APS on the immune responses of porcine pulmonary MVECs to HP-PRRSV, the effects of APS on the expression of IFIT3 and IFIT5 were investigated. According to the results of 3.4, two downregulated time points and one upregulated time point were selected to detect the IFIT3 and IFIT5 expression. The results showed that compared to the infection control group at three time points, the expression of *IFIT3* and *IFIT5* was significantly enhanced by the 100 μg/mL APS treatment at the mRNA and protein levels (Figure 8).

### 3.7. APS Inhibit HP-PRRSV Multiplication in Porcine Pulmonary MVECs

APS could increase the expression of IFIT3 and IFIT5, and then they should be able to inhibit the HP-PRRSV multiplication in porcine pulmonary MVECs. In view of this, the PRRSV N gene expression in HP-PRRSV-infected MVECs after they were treated with APS for 12, 24, and 48 h. The results showed that the expression of PRRSV N mRNA in APS-treated groups was lower than that in the infection control groups, and there was a significant difference between them at 24 and 48 hpi (Figure 9).

### 3.8. Gene Silence of IFIT3 and IFIT5 Lowers the Anti-HP-PRRSV Action of APS

To further confirm the effects of IFIT3 and IFIT5 on the anti-HP-PRRSV multiplication of APS, they were silenced in porcine pulmonary MVECs, which were then treated with 100 μg/mL APS for 24 h, and the HP-PRRSV multiplication was detected. As shown in Figure 10, compared to the expression of PRRSV N in the APS-treated MVECs at both the mRNA and protein levels, it was significantly higher in the gene-silenced MVECs. On the other hand, when using the gene-silenced MVECs, the APS treatment significantly lowered the expression of PRRSV N protein. The results suggested that the gene silence of *IFIT3* or *IFIT5* significantly weakened the inhibitory action of APS on the HP-PRRSV multiplication, which, combined with the results of 3.6, show that APS play the anti-HP-PRRSV effect at least partly by upregulating the expression of IFIT3 and IFIT5.

## 4. Discussion

In this study, we investigated the replication characteristics of HP-PRRSV in porcine pulmonary MVECs, the ultrastructural change characteristics of HP-PRRSV-infected porcine pulmonary MVECs, and the kinetic characteristics of *IFN-α*, *IFIT3*, and *IFIT5* expression. The results showed that the innate immune response of porcine pulmonary MVECs was triggered immediately after the HP-PRRSV infection, which made the viral load decrease within 24 h. However, the production of *IFN-α*, *IFIT3*, and *IFIT5* underwent a period of decline until 24 hpi after a temporary increase. Although their expression was upregulated from 48 hpi, the viruses had gained good adaptation to porcine pulmonary MVECs, and their lower expression was insufficient to inhibit the multiplication of HP-PRRSV, which resulted in an increase in the viral load in MVECs and their ultrastructural lesions. The results suggest that it is essential to clear the infected virus by an innate immune response and that the regulation of MVECs should be performed before infection or in the early period of infection to promote the production of *IFN-α*, *IFIT3*, and *IFIT5*, which was verified in effects of the APS treatment on the multiplication of HP-PRRSV.

Innate immunity is the first line of defense against pathogen invasion, and type I interferon is the most important member of its effectors. HP-PRRSV is sensitive to type I interferon, and it can induce interferon suppression or delay, as demonstrated in vitro and in vivo experiments [20,21,22], which shows its inhibitory effect on the innate immune response of the host. In this study, the expression of *IFN-α* in porcine pulmonary MVECs decreased over a period of time after the HP-PRRSV infection, which also suggests that the HP-PRRSV HN strain can inhibit the innate immune response of porcine pulmonary MVECs. The classical signaling pathway of type I interferon can induce the expression of hundreds of ISGs, including IFIT3 and IFIT5 [23]. Our study showed that the expression of IFIT3 and IFIT5 induced by HP-PRRSV displayed the same kinetics as that of *IFN-α*. Notably, the initial temporary upregulation of *IFN-α*, *IFIT3*, and *IFIT5* in MVECs was observed, which may be attributed to the stress reaction of the innate immune response in MVECs to HP-PRRSV at the early stage of infection. However, in order to survive and proliferate, HP-PRRSV would regulate the innate immune response of host cells and influence their immune cytokine response [24], which may explain why the expression of *IFN-α*, *IFIT3*, and *IFIT5* in porcine pulmonary MVECs decreased at 6, 12, and 24 hpi. However, the mechanism of its subsequent increase at 48 and 72 hpi deserves further research.

IFIT family members can block multiple stages of the viral replication to achieve antiviral actions and regulate the body’s innate immune response through various pathways. Previous studies showed that overexpression of IFIT3 in Marc 145 cells decreases the HP-PRRSV replication and that gene silencing of IFIT3 increases its infection [10]. IFIT3 could also enhance IRF3-mediated gene expression to improve the innate immune response by interacting with TBK1 and MAVS on mitochondria, which has been proven to be an important innate immunomodulator [25]. Similarly, IFIT5 could increase the antiviral effect by enhancing innate immune signaling pathways such as NF-κB [26]. Our results showed that the viral load significantly increased in porcine pulmonary MVECs after silencing the genes of *IFIT3* and *IFIT5*, which suggest that they could inhibit HP-PRRSV replication. Moreover, the silence of *IFIT5* showed a stronger effect on the HP-PRRSV replication than *IFIT3*. It is the first time to study the innate immune response of porcine pulmonary MVECs to HP-PRRSV. Considering the critical anatomical position of MVECs in vivo, the suppression of their innate immune response would contribute to the spread of HP-PRRSV to other tissues and organs. In addition, the dysfunction of MVECs would affect the transendothelial migration of immune cells in the blood [27] and then disturb the acquired immune response to HP-PRRSV.

APS are the most commonly used herbal polysaccharide to enhance immune response and can improve the expression of various immune molecules [28]. However, their immunoregulatory activities were mainly studied from classical immune cells [29,30], and particularly their regulation of the immune response to HP-PRRSV has not been investigated from the perspective of MVECs. Although the effects of APS on the production of type I interferons have been well documented, the role of IFITs in their actions has not been focused on. Our study demonstrated that the inhibitory effect of APS on the HP-PRRSV multiplication at least partially depends on the presence of IFIT3 and IFIT5. Moreover, APS could significantly upregulate the expression of IFIT3 and IFIT5 regardless of the effects of HP-PRRSV on them. In view of the early expression decline of IFIT3 and IFIT5 after the HP-PRRSV infection and the significant increase in virus multiplication after 72 h in porcine pulmonary MVECs, we suggest that APS should be administered to the infected piglets as soon as possible. However, the kinetic characteristics of APS action need to be further studied.

## 5. Conclusions

In conclusion, the effects of the HP-PRRSV infection on the expression of IFIT3 and IFIT5 in porcine pulmonary MVECs are distinct at different infection stages, IFIT3 and IFIT5 play an important role in inhibiting the HP-PRRSV replication, and APS suppress the multiplication of HP-PRRSV by upregulating their expression.

## Figures and Tables

**Figure 1 viruses-14-01919-f001:**
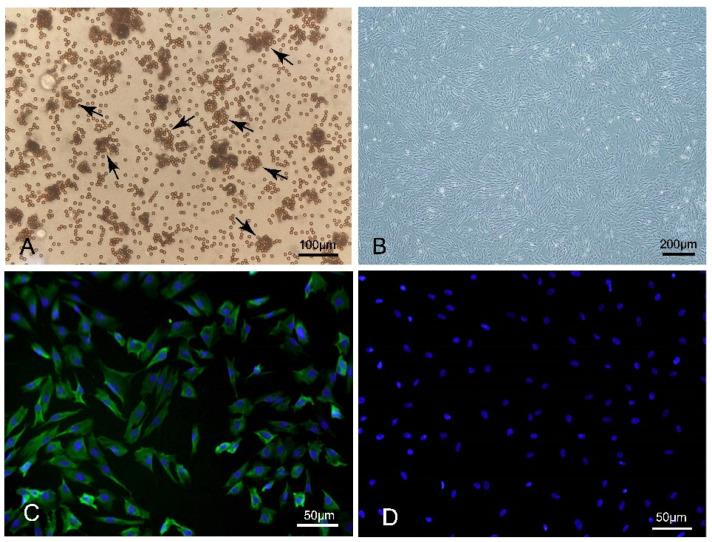
Microscopic morphology and CD31 immunofluorescence staining of porcine pulmonary MVECs. (**A**) Porcine pulmonary MVECs conjugated with CD31 immunomagnetic beads (**B**) Sub-confluent MVECs. (**C**) Positive staining for CD31. (**D**) Negative control of CD31 immunofluorescence staining.

**Figure 2 viruses-14-01919-f002:**
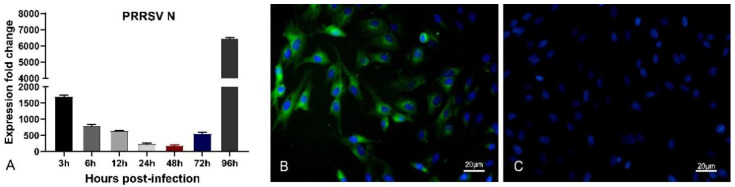
Determination of PRRSV N in porcine pulmonary MVECs. (**A**) The expression of PRRSV N mRNA was detected by quantitative RT-PCR from 3 hpi to 96 hpi. (**B**) Immunofluorescence staining of PRRSV N protein at 24 hpi. (**C**) Negative control to PRRSV N protein. Data are mean ± SD from three independent experiments. Statistical analysis was performed by Student’s *t*-test.

**Figure 3 viruses-14-01919-f003:**
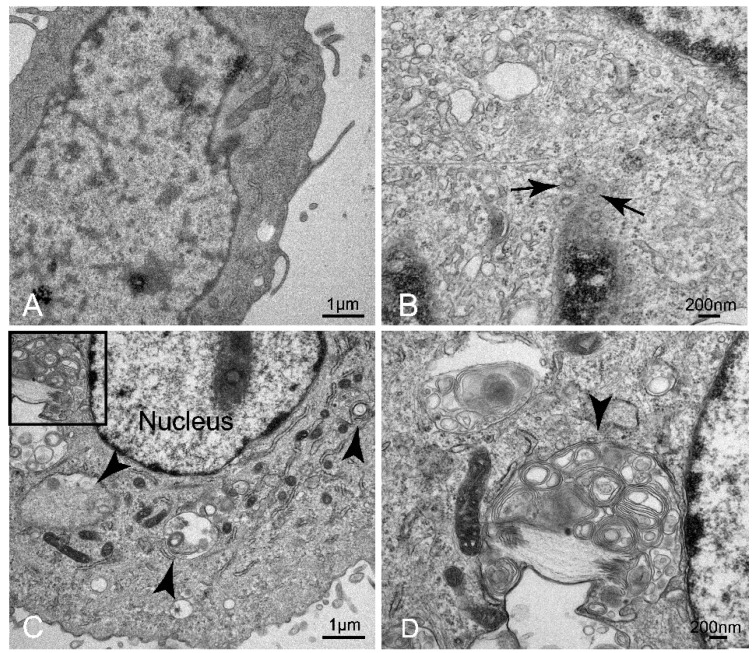
Ultrastructure analysis of HP-PRRSV-infected porcine pulmonary MVECs. (**A**) Normal control group. (**B**–**D**) HP-PRRSV-infected group. (**D**) is the enlarged insert from (**C**) in the solid line rectangular frame. Black arrows indicate virus particles at 24 hpi. (**B**) Arrowheads indicate autophagosomes at 72 hpi.

**Figure 4 viruses-14-01919-f004:**
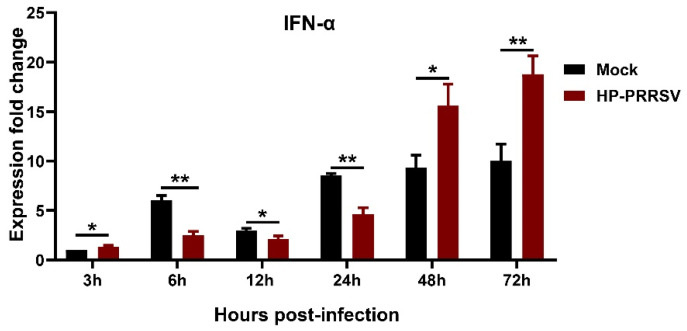
Kinetics of the mRNA expression of *IFN-α* in HP-PRRSV-infected porcine pulmonary MVECs. MVECs were infected with the HP-PRRSV HN strain, and total mRNA was extracted at 3, 6, 12, 24, 48, and 72 hpi. Its gene expression was analyzed by quantitative RT-PCR. Data are mean ± SD from three independent experiments. Statistical analysis was performed by Student’s *t*-test. * *p* < 0.05; ** *p* < 0.01.

**Figure 5 viruses-14-01919-f005:**
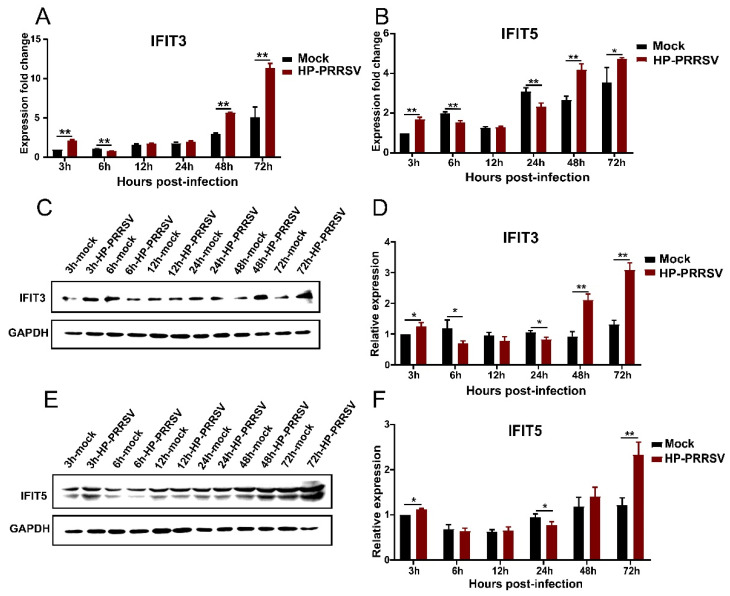
Effects of HP-PRRSV infection on the expression of IFIT3 and IFIT5 in porcine pulmonary MVECs. Porcine pulmonary MVECs were infected with the HP-PRRSV HN strain, and total cellular RNA or protein was extracted at 3, 6, 12, 24, 48, and 72 hpi. (**A**,**B**) The expression of *IFIT3* and *IFIT5* was detected by quantitative RT-PCR. (**C**,**E**) The expression of IFIT3 and IFIT5 protein was detected by Western blotting. (**D**,**F**) The gray values of the IFIT3 and IFIT5 protein bands were analyzed by ImageJ software and normalized to that of the control groups at 3 hpi. Data are mean ± SD from three independent experiments. Statistical analysis was performed by Student’s *t*-test. * *p* < 0.05; ** *p* < 0.01.

**Figure 6 viruses-14-01919-f006:**
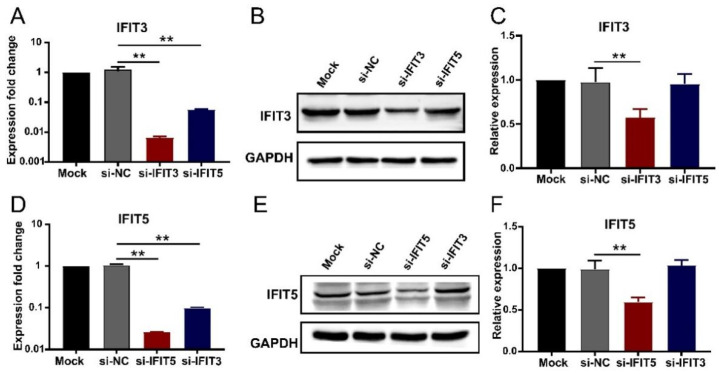
Effects of siRNA transfection on the expression of IFIT3 and IFIT5 in porcine pulmonary MVECs. Porcine pulmonary MVECs were transfected with siRNA of Negative Control (NC), *IFIT3*, or *IFIT5* for 36 h. (**A**,**D**) The total cellular RNA was extracted for detection of the *IFIT3* and *IFIT5* expression by quantitative RT-PCR. (**B**,**E**) The total cellular protein was extracted for detection of IFIT3 and IFIT5 protein by Western blotting. (**C**,**F**) The gray values of the IFIT3 and IFIT5 protein bands were analyzed by ImageJ software and normalized to that of the negative control group. Data are mean ± SD from three independent experiments. Statistical analysis was performed by Student’s *t*-test. ** *p* < 0.01.

**Figure 7 viruses-14-01919-f007:**
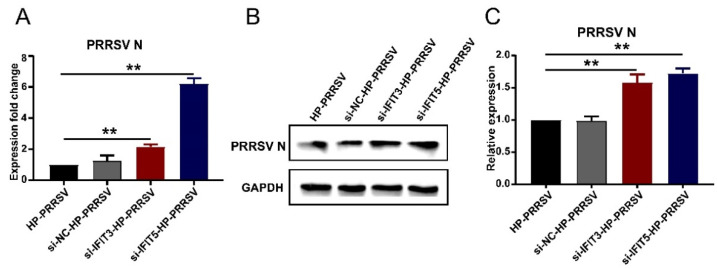
Effects of silencing IFI3 and IFIT5 expression on the HP-PRRSV multiplication. Porcine pulmonary MVECs were transfected with siRNA of *IFIT3* and *IFIT5* or Negative Control (NC) for 36 h and then infected with the HP-PRRSV HN strain for 24 h. (**A**) The expression of PRRSV N mRNA was detected by quantitative RT-PCR, (**B**)and its protein was detected by Western blotting. (**C**) The gray values of the PRRSV N protein bands were analyzed by ImageJ software and normalized to the control group. Data are mean ± SD from three independent experiments. Statistical analysis was performed by Student’s *t*-test. ** *p* < 0.01.

**Figure 8 viruses-14-01919-f008:**
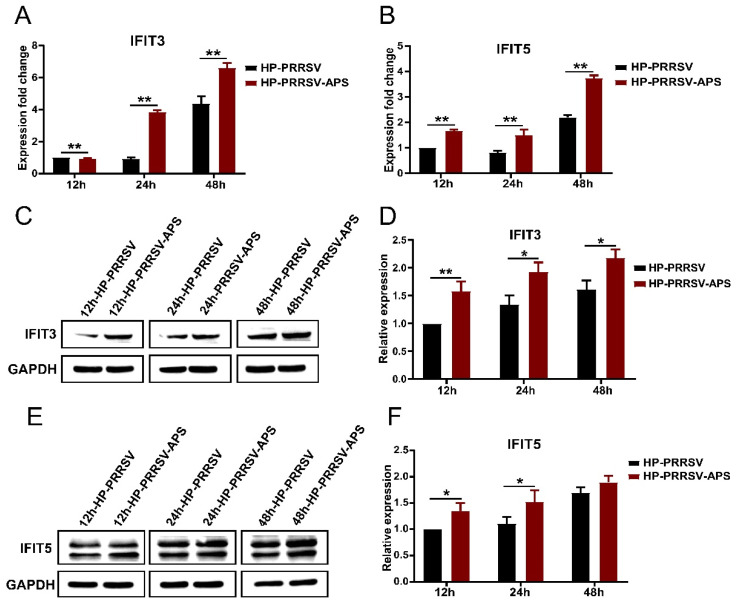
Effects of APS on the expression of IFIT3 and IFIT5 in HP-PRRSV-infected MVECs. Porcine pulmonary MVECs were infected with the HP-PRRSV HN strain for 1 h and then incubated in the maintenance medium containing 100 μg/mL APS. (**A**,**B**) The total cellular RNA was extracted at 12, 24, and 48 hpi for the detection of *IFIT3* and *IFIT5* by quantitative RT-PCR. (**C**,**E**) The total cellular protein was extracted at 12, 24, and 48 hpi for the detection of IFIT3 and IFIT5 by Western blotting. (**D**,**F**) The gray values of the IFIT3 and IFIT5 protein bands were analyzed by ImageJ software and normalized to the 12h virus control group. Data are mean ± SD from three independent experiments. Statistical analysis was performed by Student’s *t*-test. * *p* < 0.05; ** *p* < 0.01.

**Figure 9 viruses-14-01919-f009:**
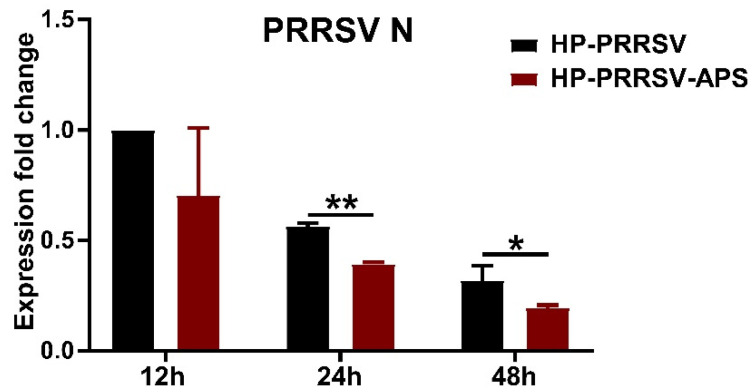
Effects of APS on HP-PRRSV multiplication in porcine pulmonary MVECs. Porcine pulmonary MVECs were infected with the HP-PRRSV HN strain and then incubated in the maintenance medium containing 100 μg/mL APS. The total cellular RNA was extracted at 12, 24, and 48 hpi for the detection of PRRSV N mRNA by quantitative RT-PCR. Data are mean ± SD from three independent experiments. Statistical analysis was performed by Student’s *t*-test. * *p* < 0.05; ** *p* < 0.01.

**Figure 10 viruses-14-01919-f010:**
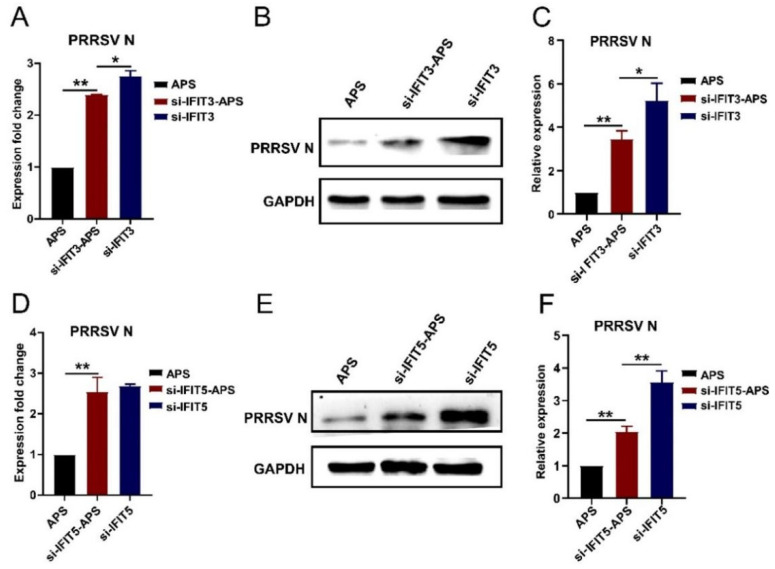
Effects of silencing IFIT3 and IFIT5 on the anti-HP-PRRSV action of APS. Porcine pulmonary MVECs were infected with the HP-PRRSV HN strain and incubated in the maintenance medium with 100 μg/mL APS for 24 h after silencing the *IFIT3* or *IFIT5* gene. (**A**,**D**) The total cellular RNA was extracted for the detection of PRRSV N mRNA by quantitative RT-PCR. (**B**,**E**) The total cellular protein was extracted for the detection of PRRSV N protein by Western blotting. (**C**,**F**) The gray values of the PRRSV N protein bands were analyzed by ImageJ software and normalized to the APS group. Data are mean ± SD from three independent experiments. Statistical analysis was performed by Student’s *t*-test. * *p* < 0.05; ** *p* < 0.01.

## Data Availability

The datasets generated for this study are available on request to the corresponding author.

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
