# Peer review of "IFIT3 and IFIT5 Play Potential Roles in Innate Immune Response of Porcine Pulmonary Microvascular Endothelial Cells to Highly Pathogenic Porcine Reproductive and Respiratory Syndrome Virus"

_viruses, 2022, doi:10.3390/v14091919_

Round 1
Reviewer 1 Report
In this manuscript, the authors investigated the effect of PRRSV on the expression of type I Interferon, IFIT3, and IFIT5 in MVECs. Then, they studied the mechanisms about how APS inhibited PRRSV.
Major concerns:
1. The authors used MVECs infection model to do the whole experiments. However, their data in Figure 2 are not convincing that PRRSV can infect MVECs. From this figure, my conclusion is that PRRSV does not infect MVECs. They must do more experiments to confirm their conclusion.
2. In figure 1, the authors need more controls to make sure they get the right cells.
Other concerns:
1. Since they used highly pathogenic PRRSV to the experiments, they need to describe the highly pathogenic PRRS in introduction (not about the classical PRRSV).
2. The English needs to be improved.
Author Response
Dear reviewer,
Thank you for reviewing our manuscript and for the constructive comments, which greatly helped us to improve this manuscript. We have carried out a comprehensive revision on the manuscript in accordance to your comments and all corrections were marked using "track changes" on our revised manuscript. And point-by-point responses to the comments were as follows:
Major concerns:
- The authors used MVECs infection model to do the whole experiments. However, their data in Figure 2 are not convincing that PRRSV can infect MVECs. From this figure, my conclusion is that PRRSV does not infect MVECs. They must do more experiments to confirm their conclusion.
Response: We appreciate the reviewer’s strict and prudent attitude. Indeed, the reliable MVECs infection model is the foundation of our study. Therefore, the susceptibility of porcine pulmonary MVECs to HP-PRRSV has been specially studied before through the CPE assays, viral antigen detection, the infectivity of released viruses, etc, and the results have been published (Li P, et al. Susceptibility of porcine pulmonary microvascular endothelial cells to porcine reproductive and respiratory syndrome virus. J Vet Med Sci. 2020, 82(9): 1404-1409). In this manuscript, the infection of HP-PRRSV in porcine pulmonary MECs were reconfirmed mainly by the TEM observation of the virus particles and their ultrastructural lesions. As for Figure 2, we would like to mainly know the virus replication kinetics from it and compare it to the production kinetics of IFN-α, IFIT3 and IFIT5. Maybe the continuous decline of the virus titers from 3 to 48 hpi brought the reviewer distrust. Actually, the expression of PRRSV N mRNA significantly increased from 72 h after infection, which was also demonstrated in the previous article by Li P, et al. To better show the increase of HP-PRRSV multiplication, we supplemented the qPCR datum at 96 hpi and the immunofluorescence image of PRRSV N protein in Figure 2.
- In figure 1, the authors need more controls to make sure they get the right cells.
Response: Thank you very much for your suggestion. We understand your concern as using the correct and pure cells is very important for this study. The manual isolation and purification of MVECs is a challenging technology. However, we have established an immunomagnetic method for separating MVECs using an automatic magnetic separation system, which has been described in detail in our previous publication (Wu Y, et al. Isolation and culture of rat intestinal mucosal microvascular endothelial cells using immunomagnetic beads. J Immunol Methods. 2022, 507: 113296). The porcine pulmonary MVECs were purified by CD31 immunomagnetic beads method from the primary cell culture preliminarily isolated by Type Ⅱ collagenase digestion and differential adhesion method, which have been applied successfully more than 5 years in our lab and provides a guarantee for getting the right MVECs. In order to prove the reliability of our isolation and purification method, the photo of CD31+ positive cells binding beads were supplemented in Figure 1.
Other concerns:
- Since they used highly pathogenic PRRSV to the experiments, they need to describe the highly pathogenic PRRS in introduction (not about the classical PRRSV).
Response: We appreciate the reviewer’s suggestion. We describe the highly pathogenic PRRSV in the introduction and the relevant references were replaced with.
- The English needs to be improved.
Response: We are very sorry for our poor English. Our manuscript was revised throughout and then polished by a native English speaker.
Other responses
- References in the manuscript were revised.
- The parts of introduction and discussion were revised of the manuscript, such as the expression and grammar, etc.
- The label "PRRSV" in Figure 4, 5, 7, 8 and 9 was changed to "HP-PRRSV", and the "dotted boxes" in Figure 3C were changed to " arrowheads".
4. Reference 31 was deleted.

Reviewer 2 Report
Line 215-217 left me confused. The figure 4 does not show mRNA elevation above the control until 48 hours. The expression relative to control maybe suppressed early but I'd want to see it repeated before I would assume anything but a statistical anomaly. It was up relative to base line but less than control so I might argue it was suppressed by viral infection early. Perhaps mention of an putative initial rise which did occur but was less than the control is unwarranted. line 344 and 345 of discussion also talks about early increase.
Line 231 the expression of MVEC's is incorrect it was expression of IFIT in those cells so this need correction. Fig 5 is nice the mRNA and protein data support each other.
Line 273 Fig 7 the choice of timing seem odd. The first studied demonstrate the IFIT production was at 48-72 hours and that seems more appropriate to evaluate viral replication effect here? Why is there PRRS N in the mock cells? Also it is not clear looking at figures 6 and 7 what exactly Mock and Si NC represent as it is not mentioned in legend or text. that should be clarified to facilitate reading the manuscript by a broad audience.
Figure 8 and 9 labels PRRS and APS are misleading as it was PRRS alone and PRRS plus APS think about clarifying this.
Author Response
Dear reviewer,
We appreciate very much your comments and suggestions, which are very helpful for us to improve this manuscript. We have carried out a comprehensive revision on the manuscript in accordance to your comments. The revisions were marked in red using "track changes" in the manuscript and addressed point by point below.
- Line 215-217 left me confused. The figure 4 does not show mRNA elevation above the control until 48 hours. The expression relative to control maybe suppressed early but I'd want to see it repeated before I would assume anything but a statistical anomaly. It was up relative to base line but less than control so I might argue it was suppressed by viral infection early. Perhaps mention of an putative initial rise which did occur but was less than the control is unwarranted. line 344 and 345 of discussion also talks about early increase.
Response: We appreciate the reviewer’s strict and prudent attitude. Actually we had the same doubt as the reviewer at first, because its kinetics characteristic was out of the ordinary perception. However, the similar results were gained from several repeated experiments. So we believe that it can not be attributed to a statistical anomaly, although we can’t provide a definite explanation for the results now. So far, the important role of MVECs in innate and acquired immunity has been demonstrated, especially they prevent the virus from spreading to the whole body by the innate immune response. However, it is known that many strategies of interfere with the immune response are evolved by HP-PRRSV, and there is complicated competition between the antiviral capability of host cells and anti-immune response ability of HP-PRRSV. Therefore, we speculate that porcine pulmonary MVECs initially displayed a stress reaction to the HP-PRRSV infection, and then the virus induced the decrease of IFITs to suppress the innate immune response of MVECs, which probably recovered again after a period time. The speculation was also described in line of Discussion. Its exact mechanisms deserve further studies.
- Line 231 the expression of MVEC's is incorrect it was expression of IFIT in those cells so this need correction. Fig 5 is nice the mRNA and protein data support each other.
Response: We apologize for our negligence and are grateful for your comments. “the expression of MVECs” should be “the expression of IFIT3 and IFIT5”. We have made the correction in the manuscript and marked the revisions using "track changes".
- Line 273 Fig 7 the choice of timing seem odd. The first studied demonstrate the IFIT production was at 48-72 hours and that seems more appropriate to evaluate viral replication effect here? Why is there PRRS N in the mock cells? Also it is not clear looking at figures 6 and 7 what exactly Mock and Si NC represent as it is not mentioned in legend or text. that should be clarified to facilitate reading the manuscript by a broad audience.
Response: We appreciate the reviewer’s comments, and your comments are of constructive help to our manuscript. (1) Considering the IFIT production, the more appropriate time point was really 48-72 hours. However, the time points of subsequent experiments included 12 h, 24 h, and 48 h, and we aimed to preliminarily evaluate at the medium time point at first. (2) The group label of “HP-PRRSV” was marked as “Mock” by mistake, and we revised all the group labels in the Figure 4 and uploaded the updated figure. Furthermore, the group labels were also annotated in the figure legends. (3) The group labels in Figure 6 and 7 were revised and annotated in the figure legends.
- Figure 8 and 9 labels PRRS and APS are misleading as it was PRRS alone and PRRS plus APS think about clarifying this.
Response: We apologize for our unclear group labels in Figure 8 and 9, and thanks for the reviewer's comment. The labels in Figures 8 and 9 were changed and were further annotated in the figure legends.
Other responses
- References in the manuscript were revised.
- The parts of abstract, introduction and discussion were revised of the manuscript,such as the expression and grammar, etc.
- The label "PRRSV" in Figure 4, 5, 7, 8 and 9 is changed to "HP-PRRSV", and the "dotted boxes" in Figure 3C were changed to " arrowheads".
- The PCR and immunofluorescence tests of PRRSV N were supplemented in "3.2. Confirmation of porcine pulmonary MVECs infected with HP-PRRSV", and the revised results can be found in the revised manuscript.
- Reference 31 was deleted.
- The photo of CD31+ positive cells binding beads were supplemented in Figure 1.
